# Pacific Ocean Neutrino Experiment

**Paweł Malecki** on behalf of the P-ONE Collaboration

H. Niewodniczański Institute of Nuclear Physics, Polish Academy of Sciences, ul. Radzikowskiego 152, 31-342 Krakow, Poland; pawel.malecki@ifj.edu.pl

**Abstract:** Following the breakthrough discoveries of very-high-energy neutrinos of astrophysical origin by IceCube, a new field of research, neutrino astronomy, was established in the previous decade. Even though two extragalactic point sources of such neutrinos have been identified by now, TXS 0506+056 and NGC 1068, the origin and processes of the creation of astrophysical neutrinos are still mostly unexplored. To advance quickly in this new field, more neutrino telescopes are needed. This article describes the current status and plans for the development of the Pacific Ocean Neutrino Experiment (P-ONE), which is under construction in the Pacific Ocean near Vancouver Island. The deployment of P-ONE is expected to start in 2025, exploiting the already available deep-sea infrastructure provided by Ocean Networks Canada. P-ONE will complement the existing IceCube, Baikal-GVD, and KM3NeT neutrino telescopes not only with its large detection volume, but also by providing insight into the southern celestial hemisphere, where the central region of the Galactic Plane is located.

**Keywords:** neutrino; astronomy; very-high energy; Water Cherenkov telescope; Pacific Ocean

## 1. Introduction

*Neutrino astronomy* is a novel field of research that has been firmly established thanks to the discoveries of IceCube [1] in the previous decade. They include the first observations of very-high-energy astrophysical neutrinos [2,3], hints of association of their flux with blazars [4], as well as the discovery of their first point source in the sky [5], the TXS 0506+056 blazar, followed by a similar hint for the Seyfert galaxy NGC 1068 [6]. After over 10 years of successful operation of the IceCube detector, it is, however, clear that in order to keep the pace of the development in the field, a significant increase in both the accessible event rate and sky coverage is needed. While a huge increase in the IceCube-instrumented volume is expected [7], a better view on the Southern Sky is also needed [8], which implies the need to build neutrino telescopes in the Northern Hemisphere to complement IceCube and efficiently observe neutrinos traversing the Earth, ensuring a cutoff of the atmospheric muon background. While Baikal-GVD [9] and KM3NeT [10] have long been under development in the Baikal lake and Mediterranean sea, respectively, and, in addition, a new project, TRIDENT [11], is planned in the South China Sea, the Pacific Ocean Neutrino Experiment (P-ONE) [12] is in the advanced construction phase and is about to be deployed in the Pacific Ocean, West of Vancouver Island, starting in 2025. It will complement the sky coverage of the existing telescopes, see Figure 1. Moreover, because the scattering length in water is much longer than in ice, it will also show a much better angular resolution for incoming neutrinos (with respect to IceCube) and help in identifying sources in the sky.

P-ONE, located in the Northern Hemisphere, will provide data complementary to those of IceCube by extending the coverage of the southern sky.[1] Its physics goals cover the search for neutrinos from known astrophysical objects and the detection of unidentified sources in the sky. The potential galactic sources are likely located mainly in the vicinity of the Galactic Center and in the Galactic Plane and include supernova remnants, pulsars, the neighborhood of the black hole *Sgr A\**, and many others. The energy spectrum of galactic neutrinos fills the energy range $10^3$–$10^6$ GeV, according to production models.

On the other hand, extra-galactic objects—Active Galactic Nuclei, Gamma-Ray Bursts, starburst galaxies, and galaxy clusters—could generate neutrinos in the energy range of $10^4$–$10^8$ GeV or higher. P-ONE will also contribute to multi-messenger astronomy where combined studies of chosen events with other detectors may lead to a higher significance of the results. Another direction is to investigate the characteristics of the diffuse neutrino flux at energies above $10^4$ GeV. Searches for Dark Matter particle decays are also possible with the P-ONE telescope. Tau-neutrino tagging is also going to be possible in P-ONE as well as muon bundle discrimination.

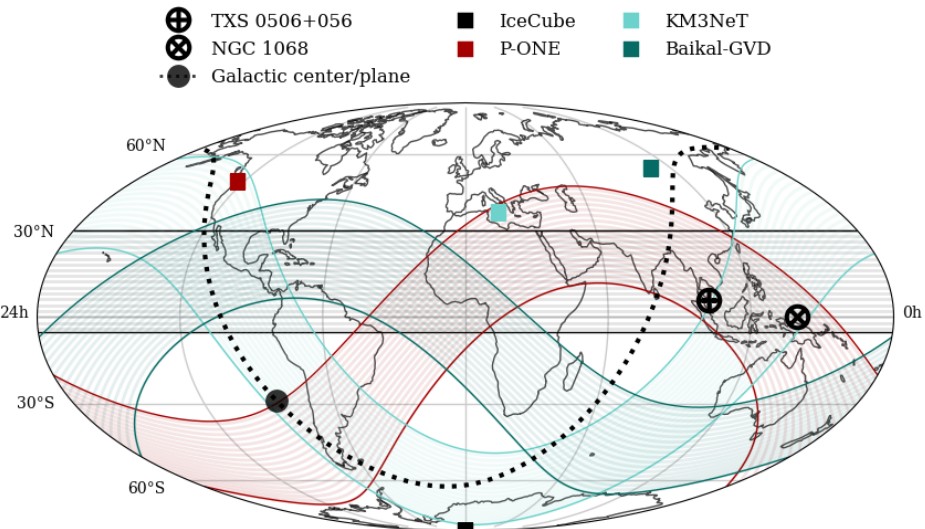

**Figure 1.** The sky coverage of the existing and planned neutrino telescopes. The bands show sky coverage of each experiment in terms of maximum discovery potential for sources with energy spectrum $E^{-2}$. At the boundaries, the discovery potential drops by a factor of 2. From Ref. [13].

## 2. Experimental Site

As neutrinos interact only weakly and their flux drops with energy as $E^{-2} - E^{-3}$, very large instrumented volumes, of the size of a cubic kilometer, are needed to observe the astrophysical neutrino flux. Construction of such detectors from scratch is not possible, but instrumentation of the existing large reservoirs of transparent matter, such as antarctic ice (IceCube) or water (Baikal-GVD, KM3NeT, P-ONE) can be exploited. Incoming neutrinos will sometimes undergo interaction in which secondary charged particles are produced. They can be subsequently detected thanks to the Cherenkov radiation, where the secondary particles emit within or in the vicinity of the instrumented volume.

The site chosen for the construction of P-ONE is Cascadia Basin in the Pacific Ocean, about 600 km West of Vancouver, Canada. It offers stable operating conditions with sufficient depth (around 2600 m), low currents, a nearly constant 2 °C temperature, and good optical properties already explored with the STRAW pathfinder missions (see Section 4). Its most unique feature, however, is the existence of a robust undersea infrastructure that is ready for use for the installation of the detector: Ocean Networks Canada (ONC) [14] hosts the NEPTUNE (North East Pacific Time-series Underwater Networked Experiment) Observatory, one of the world's most advanced ocean observatories, with an 800 km loop of cables providing power and fiber-optic data connection for the new neutrino telescope. The installation of the detector within the NEPTUNE infrastructure will allow for sequential deployment, easy maintenance, and support from ONC on marine operations. The map of the NEPTUNE observatory with the location of P-ONE is presented in Figure 2.

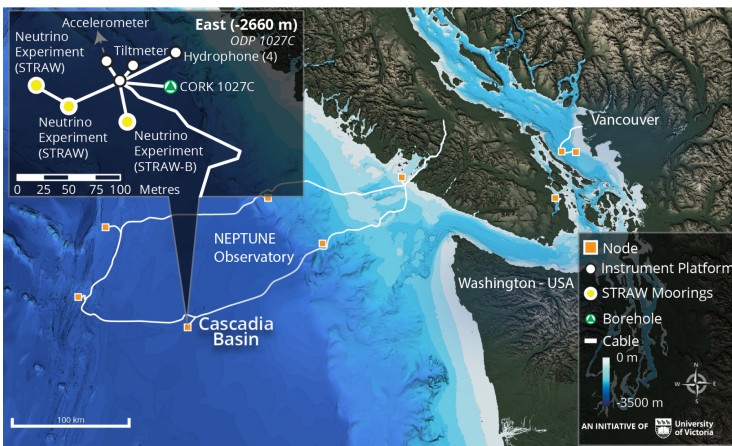

**Figure 2.** Map of ONC's NEPTUNE observatory. The two STRAW lines are located in the Cascadia Basin at a depth of 2660 m. Image courtesy of ONC.

## 3. Detector Hardware and Layout

The optical attenuation length of the ocean water is much lower than that of Antarctic ice (20–40 m vs. more than 100 m); therefore, a uniform array structure would generate very high costs. Instead, a segmented structure, similar to that of Baikal-GVD, was chosen. The individual measurement units, Precision Optical Modules (P-OM), will be connected to vertical strings, where each string will host 20 such OMs and several calibration modules (P-CAL) and will be anchored to the bottom of the ocean. A group of 10 strings will form a *cluster* comprising a cylindrical structure with a radius of 200 m and 80 m distances between strings. The vertical separation between the OMs will be 50 m. Seven such clusters are planned for the first cubic kilometer of volume to be covered. The planned detector layout is depicted in Figure 3, whereas the single P-OM and P-CAL hemispheres are visualized in Figure 4. It has to be noted that this is a proposed baseline layout and it will be optimized in the construction process to maximize the effective area for neutrino detection and to obtain the best possible angular and energy resolution, while keeping the construction costs at reasonable level.

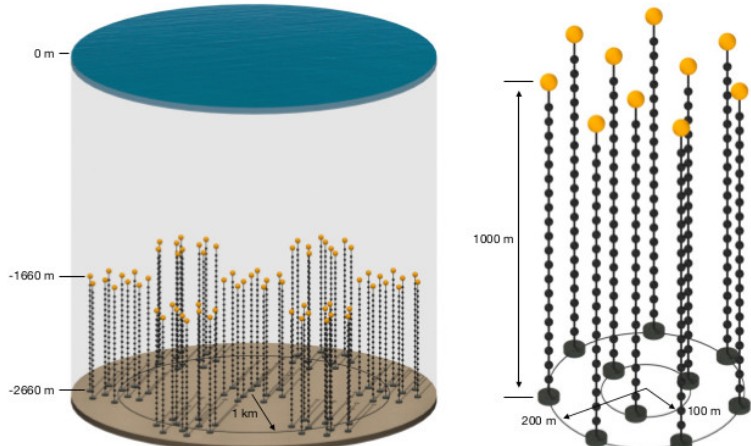

**Figure 3.** The layout of the P-ONE detector and single cluster. From [15].

The P-OMs will be housed in 17-inch glass spheres containing 16 3-inch photomultipliers (PMT, Hamamatsu R14374-10), whereas the P-CALs will contain 10 PMTs and two LED flashers for calibration of the charge and time readout, detector positions and efficiencies.

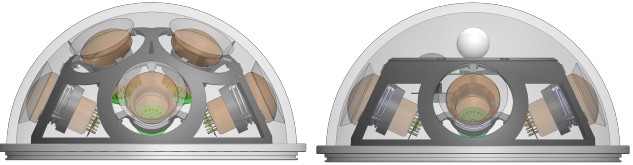

**Figure 4.** Render of P-OM hemisphere (**left**) and P-CAL hemisphere (**right**). From [16].

## 4. Pathfinder Missions and Results

The P-ONE Collaboration has developed and deployed two pathfinder missions in the experimental site, STRAW [17] and STRAW-b [18] (STRings for Absorption in Water), in 2018 and 2020, respectively, and was decommissioned after successful operation in Summer 2023. Their purpose was to explore the optical properties of the local water, measure the possible optical background, and test solutions planned for the regular measurement lines in the future.

The first pathfinder mission, STRAW, was mainly constructed to measure the attenuation length of water for wavelengths between 350 and 600 nm. It is composed of two mooring lines (or: *strings*), each containing a number of light-emitting LED flashers, the POCAMs (Precision Optical CAlibration Modules), and light-detecting sDOMs (STRAW Digital Optical Modules), each hosting two photomultipliers facing downwards and upwards, respectively. The modules are located at depths between 30 and 100 m above the sea floor and allow for the recording of light intensity from POCAMs at various distances (see Figure 5 left) [17].

The subsequent pathfinder, STRAW-b, was located 40 m away from STRAW and consisted of a single 444 m long string-hosting module enclosed in 13-inch glass-housing spheres. The modules included two LIDARs, three modules to measure environmental conditions, one muon tracker (composed of scintillator planes and SiPMs), and two spectrometer modules. The two latter ones also contained low-light cameras to register bioluminescence events in deep ocean (see Figure 5 right) [18].

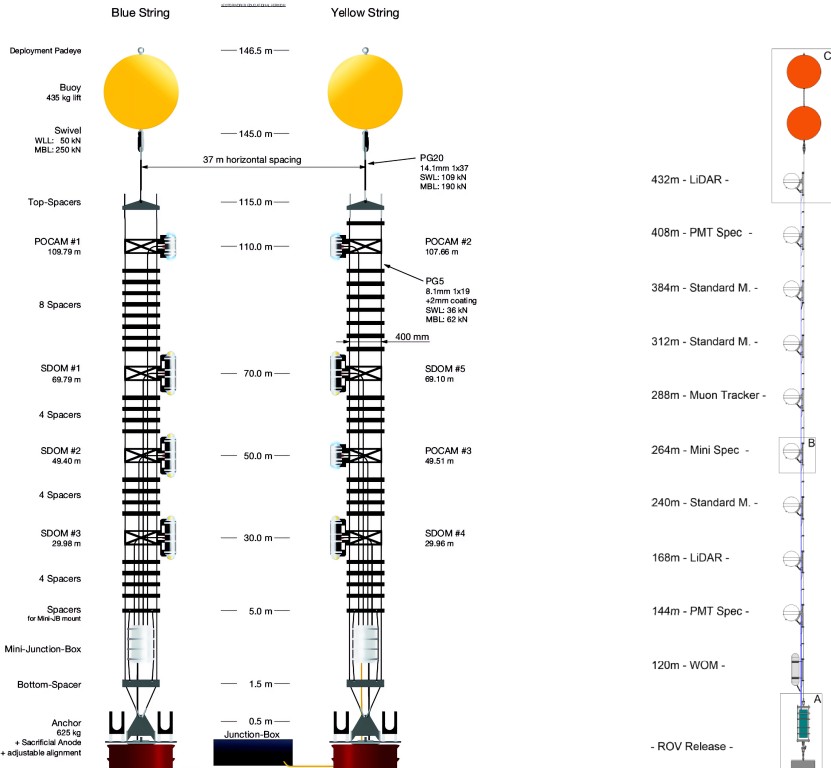

**Figure 5.** The layout of the first STRAW (**left**) and the subsequent STRAW-b (**right**).

*4.1. Results*

4.1.1. Attenuation Length

The first pathfinder provided a measurement of light attenuation length in water at the experimental site. The measurement was performed by fitting a parametric model of the whole STRAW setup to all data available at that time (2 years of exposure). The results are summarized in Figure 6. The attenuation length for the wavelength of 450 nm (closest to that of Cherenkov radiation) is $27.7^{+1.9}_{-1.3}$ m [19]. With the multi-PMT design of optical modules, this value of the attenuation length should not significantly reduce the light yield recorded by P-OMs.

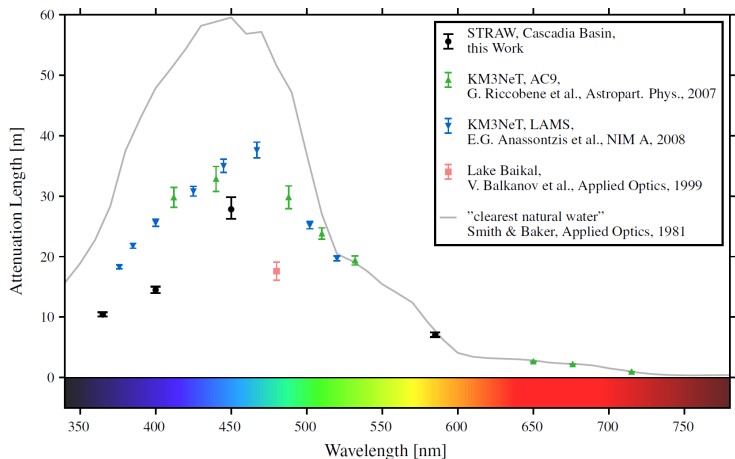

**Figure 6.** Attenuation length from STRAW measurements at various wavelengths [19]. Results from Baikal-GVD [20] and KM3NeT [21,22] and estimate for clearest ocean waters [23] are also shown for comparison.

4.1.2. Salinity and $^{40}$K Content

The presence of a Potassium-40 ($^{40}$K) radioactive isotope in the sea salt results in an ambient-light background that will be recorded by the optical sensors of P-ONE. The $\beta$ electrons from $^{40}$K decays as well as Compton-scattered electrons from excited Argon nuclei (created in the electron-capture process by $^{40}$K) will generate Cherenkov light:

$$^{40}K \rightarrow ^{40}Ca + e^- + \bar{\nu}_e, \tag{1}$$

$$^{40}K + e^- \rightarrow ^{40}Ar + \nu_e + \gamma. \tag{2}$$

Its pattern will be different from the expected future signal in the detector and will consist of coincident illumination of the top and bottom PMT in sDOMs. This measurement was performed using only photons coincident between the top and bottom sensor of the given sDOM within a narrow time window. The obtained $^{40}K$ decay rate was $78 \pm 44$ mHz. A comparison with simulations is performed to check the simulation parameters and a good agreement was found. With this result, a comparison of salinity value with results previously obtained by ONC can be performed to prove that the simulation inputs were correct. A salinity value of $2.5 \pm 1.4\%$ was found, in agreement with an ONC value of $3.482 \pm 0.001\%$ [19].

4.1.3. Background Rates from Bioluminescence

Bioluminescence is the emission of light by living organisms. This mechanism is used by many species in the deep-sea environment and may serve, e.g., for finding food, attracting mates, and evading predators. Significant light emission is expected for wavelengths of 440–500 nm; therefore, it is crucial to estimate its rate as a background for P-ONE measurements. Especially, it is important to estimate if the planned design of the data acquisition system (DAQ) is sufficient to withstand the bioluminescence background rates.

The results obtained with 2-years of exposure indicate that the maximum detection rate of 10 MHz is exceeded only very rarely, see Figure 7 (left). This is essential for the design of the P-ONE DAQ system. In addition, the fraction of time above the threshold rate is computed to estimate the background-induced dead time of the photosensors. The temporal evolution of rate percentiles is shown in Figure 7 (right), showing modulation with tidal cycles (12.5 h) [19].

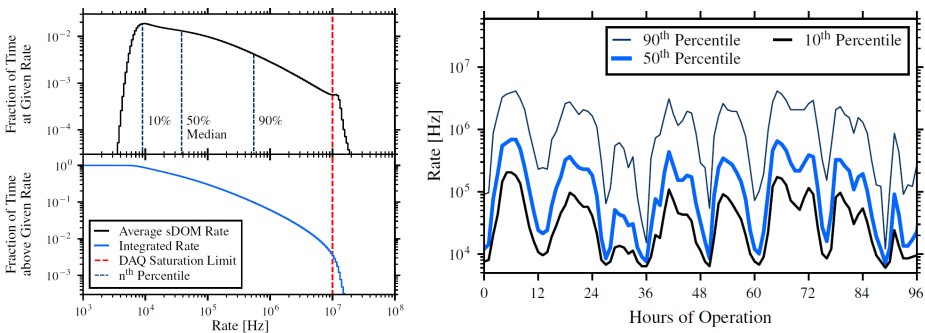

**Figure 7.** (**Left**): Single-PMT background rates. The bottom plot shows the integral fraction of time above the given rate. (**Right**): temporal evolution of background-rate percentiles [19].

### 4.1.4. Spectral Population of Bioluminescent Events

The emission spectra for many organisms are known and exhibit a characteristic wavelength thanks to the particular underlying biochemical reactions. The known spectra are cataloged [24]. The information from cameras of the STRAW-b pathfinder, in the form of RGB channels, can be transformed into *hue* or color angle. Using the water and glass-housing transmittance information, these data can then be unfolded to obtain the emission wavelengths. Figure 8 shows the resulting spectral population in addition to presenting the entries from bioluminescence catalogs. Most remarkably, the *Annelida*, with a spectral peak around 585 nm, visible only near the bottom (Camera 1), are planktonic organisms that cannot move against water currents [25].

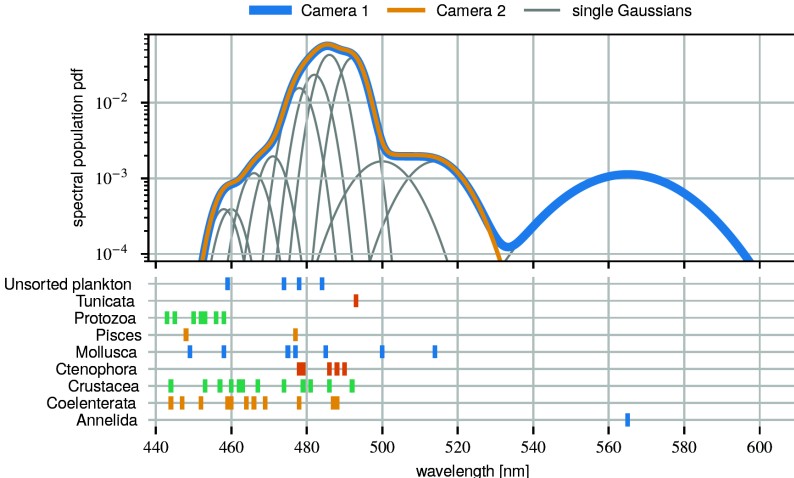

**Figure 8.** Spectral bioluminescence population to reproduce the hue distribution measured with STRAW-b cameras [25]. The lower plot shows the peak positions from organisms in a catalog [24] (colors selected for clarity).

## 5. Towards the First Measurement Line: P-ONE-1

The pathfinder missions described above pave the way for the first regular measurement line of the Telescope, namely *P-ONE-1*. It will be a 1000 m long mooring line with 20 evenly distributed optical modules, including two calibration modules, as described before. The deployment, planned for Spring 2025, will be handled by ONC. It will use a bottom–up

approach where the mooring line with Optical Modules will be contained in the mounting frame. The whole system will first be lowered to the seabed and then released by setting the float free [26].

### 6. Summary and Outlook

The Pacific Ocean Neutrino experiment is a neutrino telescope under construction in the Pacific Ocean near Vancouver island. Its planned deployment was preceded by a successful operation of the two pathfinder STRAW missions, which provided results discussed in the Sections above. The physics goal of the P-ONE experiment is to search for the flux of astrophysical neutrinos of very high energies, both in the context of the flux-properties measurement and in the searches for sources of such neutrinos in the sky. The development of the P-ONE experiment is proceeding at full speed. Especially, progress towards the construction and deployment of the first measurement line, P-ONE-1, are well advanced. The properties of the experimental site have been examined in great detail. A new neutrino telescope in the Northern Hemisphere will soon be ready to provide data complementary to those of the successful IceCube experiment, bringing Neutrino Astronomy to a new, exciting stage with extended sky coverage and detection rates to ensure its rapid progress.

**Funding:** Author's research is supported by the National Science Centre, Poland, under research project "P-ONE Experiment—Neutrino Telescope in Pacific Waters: development, calibration and early analyses", no UMO-2022/46/E/ST9/00438.

**Data Availability Statement:** Dataset available on request from the authors. Restrictions may apply.

**Acknowledgments:** We thank Ocean Networks Canada for the very successful operation of the NEPTUNE observatory, as well as the support staff from our institutions without whom P-ONE could not be operated efficiently. We acknowledge the support of Natural Sciences and Engineering Research Council, Canada Foundation for Innovation, Digital Research Alliance, and the Canada First Research Excellence Fund through the Arthur B. McDonald Canadian Astroparticle Physics Research Institute, Canada; European Research Council (ERC), European Union; Deutsche Forschungsgemeinschaft (DFG), Germany; National Science Centre, Poland; U.S. National Science Foundation-Physics Division, USA; Science and Technology Facilities Council, part of U.K. Research and Innovation, and the UCL Cosmoparticle Initiative.

**Conflicts of Interest:** The authors declare no conflicts of interest.

### Note

[1] Neutrino telescopes provide best sensitivity for up-going events—neutrinos that traverse the Earth. For such events the overwhelming background from atmospheric muons is suppressed.

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
