# Peer review of "Pacific Ocean Neutrino Experiment"

_universe, doi:10.3390/universe10020053_

Round 1

Reviewer 1 Report

Comments and Suggestions for Authors

 My comments are listed below:

1), Figure,1 it is better to change the colors of P-One and KM3NET to make a clear difference.

2), From section 4.1.2 to section 4.1.4, the description of the background should be given in general.

3), Section 6 does have a good summary of this paper. There is no description of the P-one, not even the Pathfinder.

4) In addition, the physics goal must be highlighted somehow both in the introduction and more importantly in the summary section.

Author Response

Dear Referee,

Thank you for your detailed comments. Please see my replies below.

Best regards,
Paweł Malecki

1), Figure,1 it is better to change the colors of P-One and KM3NET to make a clear difference.

-> Done, please see the updated draft. Also updated caption.

2), From section 4.1.2 to section 4.1.4, the description of the background should be given in general.

-> Done, I hope this is better now.

3), Section 6 does have a good summary of this paper. There is no description of the P-one, not even the Pathfinder.

-> Done

4) In addition, the physics goal must be highlighted somehow both in the introduction and more importantly in the summary section.
-> Done, as of introduction, it already contained a paragraph with the physics program, updated it to be more clear.

Reviewer 2 Report

Comments and Suggestions for Authors

Review of P-ONE (Malecki)The paper was prepared based on a talk by invited participants at a conference and submitted for peer review.  It is interesting as a short standalone report but covers ground already available in the literature.  If that basis is allowable for the journal, the paper is publishable. In Fig. 1, please explain the bands.  Is the sensitivity shown here for horizontal or upward-going neutrinos?  What energy range?  This figure does not come from Ref. 8.  Where does it come from?  Please make it larger for readability.What is the basis for 7 segments spaced as they are?  The original article (Ref. 11) is unclear on this as well, beyond economics.  Why 10 strings in 7 groups, and why 1 km?   Was there an optimization strategy, subject to capital costs, or how was the decision made? What is the 40K background level and count rate?  Section 4.1.2 needs to be rewritten for clarity and specificity.  It begins by discussing the 40K but ends with the sali!

 nity and no estimates of background rates.  Presumably, the salinity gives the 40K rate under some assumptions, but one cannot tell what they are. The new project in the South China Sea, TRIDENT, needs to be cited and discussed, even if briefly [Ye et al., Nature Astronomy 7, 1497 (2023)].  (Optional: what would be most relevant here is how P-ONE, Km3Net, and TRIDENT together provide new coverage beyond IceCube and ANTARES.  TRIDENT claims that scattering is less in deep ocean water (3500 m as opposed to P-ONE at 2660 m) ?why is that?  The absorption is similar.)Typos:L97: of a numberL98: Moules --> Modules

Author Response

Dear Referee,

Thank you for your detailed comments. Please see my replies below.

Best regards,
Paweł Malecki

Review of P-ONE (Malecki)The paper was prepared based on a talk by invited participants at a conference and submitted for peer review.  It is interesting as a short standalone report but covers ground already available in the literature.  If that basis is allowable for the journal, the paper is publishable.

 In Fig. 1, please explain the bands.  Is the sensitivity shown here for horizontal or upward-going neutrinos?  What energy range?  This figure does not come from Ref. 8.  Where does it come from?  Please make it larger for readability.
-> Done, updated plot, caption and description.

What is the basis for 7 segments spaced as they are?  The original article (Ref. 11) is unclear on this as well, beyond economics.  Why 10 strings in 7 groups, and why 1 km?   Was there an optimization strategy, subject to capital costs, or how was the decision made? 

-> Added a sentence about that.

What is the 40K background level and count rate?  Section 4.1.2 needs to be rewritten for clarity and specificity.  It begins by discussing the 40K but ends with the salinity and no estimates of background rates.  Presumably, the salinity gives the 40K rate under some assumptions, but one cannot tell what they are.
->Added info on 40K-related background rates.

The new project in the South China Sea, TRIDENT, needs to be cited and discussed, even if briefly [Ye et al., Nature Astronomy 7, 1497 (2023)]. 
-> Added

(Optional: what would be most relevant here is how P-ONE, Km3Net, and TRIDENT together provide new coverage beyond IceCube and ANTARES.  TRIDENT claims that scattering is less in deep ocean water (3500 m as opposed to P-ONE at 2660 m) ?why is that?  The absorption is similar.)
-> There is no such plot available yet, also I don’t know where the differences in catering length come from. As you marked this as “optional” I take the liberty of skipping it.

Typos:L97: of a numberL98: Moules --> Modules
-> Corrected

Reviewer 3 Report

Comments and Suggestions for Authors

The paper gives a good overview of the status, ideas and results of research and development for a new high-energy neutrino detector to be installed underwater in the Pacific Ocean. In general, the publication is in favour, but some questions still need to be clarified.

- abstract and intro (l8+9; l24ff): the identification of two neutrino sources are mentioned, however, meanwhile there are three sources including the Milky Way - this should be adapted in the text.

l17 + Fig 1 etc.: in principle, these kind of neutrino detectors show a 4-pi sensitivity to neutrinos. it should be shortly explained why sky-coverage by several experiments is of advantage (energy dependence?)

L41-43: the two mentioned sources are not from the Galactic Plane, however the Milky Way is identified as source - How this comes together with the statement here?

L44: from where it is known that the energy range of these neutrinos are in the TeV-PeV range?

L 51: I do not understand why IceCube and KM3NeT should not be able to identify tau-neutrinos, but P-ONE does?

L 106: what kind of device is the "muon tracker" ?

paragraphs 4.1.1 - 4.1.5: These are good results from specific measurements of important parameters for optimising the experiment and determining the sensitivity. For me as a reader, however, the results are difficult to categorise as to whether this is good or less good for P-ONE and the detection of neutrinos of astrophysical origin. One sentence per paragraph would be good here, explaining what these results ultimately mean for P-ONE. E.g. Fig6: What does the smaller attenuation length mean compared to KM3NeT in terms of angular resolution or detection efficiency?

L 168: We thank Canada sound a bit weird.....do you mean the Canadian ministry or funding agency...?

Comments on the Quality of English Language

The English is fine.

Author Response

Dear Referee,

Thank you for your detailed comments. Please see my replies below.

Best regards,
Paweł Malecki

The paper gives a good overview of the status, ideas and results of research and development for a new high-energy neutrino detector to be installed underwater in the Pacific Ocean. In general, the publication is in favour, but some questions still need to be clarified.

- abstract and intro (l8+9; l24ff): the identification of two neutrino sources are mentioned, however, meanwhile there are three sources including the Milky Way - this should be adapted in the text.
-> Done

l17 + Fig 1 etc.: in principle, these kind of neutrino detectors show a 4-pi sensitivity to neutrinos. it should be shortly explained why sky-coverage by several experiments is of advantage (energy dependence?)
-> Done

L41-43: the two mentioned sources are not from the Galactic Plane, however the Milky Way is identified as source - How this comes together with the statement here?
-> Rephrased to be more clear

L44: from where it is known that the energy range of these neutrinos are in the TeV-PeV range?
-> Added (this is from neutrino production models)

L 51: I do not understand why IceCube and KM3NeT should not be able to identify tau-neutrinos, but P-ONE does?
-> That was a bit too bold statement. Removed.

L 106: what kind of device is the "muon tracker" ?
-> Added a sentence on that

paragraphs 4.1.1 - 4.1.5: These are good results from specific measurements of important parameters for optimising the experiment and determining the sensitivity. For me as a reader, however, the results are difficult to categorise as to whether this is good or less good for P-ONE and the detection of neutrinos of astrophysical origin. One sentence per paragraph would be good here, explaining what these results ultimately mean for P-ONE. E.g. Fig6: What does the smaller attenuation length mean compared to KM3NeT in terms of angular resolution or detection efficiency?
-> Added a statement for attenuation length. As of 40K - verification of simulation parameters was done here (as stated in the text), for bioluminescence the main concern was if the event rate won’t blind the DAQ - and it won’t. 

L 168: We thank Canada sound a bit weird.....do you mean the Canadian ministry or funding agency...?
-> Thanks for catching, this was some editing error, should’ve been “Ocean Networks Canada”, corrected.

Round 2

Reviewer 1 Report

Comments and Suggestions for Authors

The authors have addressed well all the issues raised in my first review and is accepted. The manuscript is ready for publication.

Reviewer 3 Report

Comments and Suggestions for Authors

Thank the authors for clarifying the open issues. From my side the paper is ready for publication.

Comments on the Quality of English Language

fine with me